# ON THE VULNERABILITY OF DISCRETE GRAPH DIFFUSION MODELS TO BACKDOOR ATTACKS

## ABSTRACT

Diffusion models have demonstrated remarkable generative capabilities in continuous data domains such as images and videos. Recently, discrete graph diffusion models (DGDMs) have extended this success to graph generation, achieving state-of-the-art performance. However, deploying DGDMs in safety-critical applications—such as drug discovery—poses significant risks without a thorough understanding of their security vulnerabilities. In this work, we conduct the first study of backdoor attacks on DGDMs, a potent threat that manipulates both the training and generation phases of graph diffusion. We begin by formalizing the threat model and then design a backdoor attack that enables the compromised model to: 1) generate high-quality, benign graphs when the backdoor is not activated, 2) produce effective, stealthy, and persistent backdoored graphs when triggered, and 3) preserve fundamental graph properties—permutation invariance and exchangeability—even under attack. We validate 1) and 2) empirically, both with and without backdoor defenses, and support 3) through theoretical analysis.

## 1 INTRODUCTION

Diffusion models have recently driven transformative advancements in generative modeling across diverse fields: image generation Sohl-Dickstein et al. (2015); Ho et al. (2020); Dhariwal & Nichol (2021), audio generation Kong et al. (2021); Liu et al. (2023b), video generation Ho et al. (2022). Inspired by nonequilibrium thermodynamics Sohl-Dickstein et al. (2015), these models employ a unique two-stage approach involving forward and reverse diffusion processes. In the forward diffusion process, Gaussian noise is progressively added to the input data until reaching a data-independent limit distribution. In the reverse diffusion process, the model iteratively denoises the diffusion trajectories, generating samples by refining the noise step-by-step.

This success of diffusion models for *continuous* data brings new potentials for tackling graph generation, a fundamental problem in various applications such as drug discovery Li et al. (2018) and molecular and protein design Liu et al. (2018; 2023a); Gruver et al. (2024). The first type of approach Niu et al. (2020); Jo et al. (2022); Yang et al. (2023) adapts diffusion models for graphs by embedding them in a *continuous space* and adding Gaussian noise to node features and adjacency matrix. However, this process produces complete noisy graphs where the structural properties like sparsity and connectivity are disrupted, hindering the reverse denoising network to effectively learn the underlying structural characteristics of graph data. To address the limitation, the second type of approach Vignac et al. (2023); Kong et al. (2023); Chen et al. (2023b); Li et al. (2024); Gruver et al. (2024); Yi et al. (2024); Xu et al. (2024) proposes *discrete* graph diffusion model (DGDM) tailored to graph data. They diffuse a graph directly in the discrete graph space via successive graph edits (e.g., edge insertion and deletion). Especially, the recent DGDMs Vignac et al. (2023); Xu et al. (2024) can preserve the marginal distribution of node and edge types during forward diffusion and the sparsity in intermediate generated noisy graphs (more details see Section 2). In this paper, we focus on DGDMs, as they have also obtained the state-of-the-art performance on a wide range of graph generation tasks.

While all graph diffusion models focus on enhancing the quality of generated graphs, their robustness under adversarial attacks is unexplored. Adopting graph diffusion models for safety-critical tasks (e.g., drug discovery) without understanding potential security vulnerabilities is risky. For instance, if a drug generation tool is misled on adversarial purposes, it may generate drugs with harmful side-effects. We take the first step to study the robustness of DGDMs Vignac et al. (2023); Xu et al. (2024) against backdoor attacks. We note that several prior works Zhang et al. (2021); Xi et al. (2021); Yang et al. (2024) show graph *classification* models are vulnerable to backdoor

attacks. In this setting, an attacker injects a *subgraph* backdoor trigger into some training graphs and alters their labels as the attacker-chosen target label. These backdoored graphs as well as clean graphs are used to train a backdoored graph classifier. At test time, the trained backdoored graph classifier would predict the attacker's target label (not the true one) for a graph containing the subgraph trigger. *However, generalizing these attack ideas for our purpose is insufficient*: backdoor attacks on graph classifiers simply alter the training graphs and their labels to implant backdoors, while on graph diffusion models require complex alterations to not only the training graphs, but also the underlying forward and reverse diffusion processes.

**Our work:** We aim to design a backdoor attack by utilizing the unique properties of discrete noise diffusion and denoising within training and generation in DGDMs. At a high-level, the backdoored DGDM should satisfy below goals:

1. *Utility preservation:* The backdoored DGDM should minimally affect the quality of the generated graphs without activating the backdoor trigger.
2. *Backdoor effectiveness, stealthiness, and persistence:* The backdoored DGDM should generate expected backdoored graphs when the trigger is activated. Moreover, the backdoor should be stealthy and persistent, meaning not easy to be detected/removed via backdoor defenses.
3. *Permutation invariance:* Graphs are invariant to the node reorderings. This requires the learnt backdoored model should not change outputs with node permutations.
4. *Exchangeability:* All permutations of generated graphs should be equally likely Köhler et al. (2020); Xu et al. (2022). In other words, the generated graph distribution is exchangeable.

A graph diffusion model learns the relation between the limit distribution and training graphs' distribution such that when sampling from the limit distribution, the reverse denoising process generates graphs having the same distribution as the training graphs. We are motivated by this and design the attack on DGDMs to ensure: i) backdoored graphs and clean graphs produce different limit distributions under the forward diffusion process; and ii) the relations between backdoored/clean graphs and the respective backdoored/clean limit distribution are learnt after the backdoored DGDM is trained. Specifically, we use *subgraph* as a backdoor trigger, following backdoor attacks on graph classification models (Zhang et al., 2021; Xi et al., 2021; Yang et al., 2024). We then use the forward diffusion process in DGDMs for clean graphs, and *carefully design the forward diffusion process for backdoored graphs (i.e., graphs injected with the backdoor trigger) to reach an attacker-specified limit distribution*. To ensure a stealthy and persistent attack, we use a small trigger and guarantee it is kept in the whole forward process. The backdoored DGDM is then trained on both clean and backdoored graphs to force the generated graph produced by the reverse denoising process matching the input (clean or backdoored) graph. We also prove our backdoored DGDM is node permutation invariant and generates exchangeable graph distributions. Our contributions are summarized as follows.

- We are the first work to study the robustness of graph diffusion models under graph backdoor attacks. We clearly define the threat model and design the attack solution.
- We prove our backdoored graph diffusion model is *permutation invariant* and generates *exchangeable* graphs—two key properties in graph generative models.
- Evaluations on multiple molecule datasets show our attack marginally affects clean graph generation, and generates the stealthy and persistent backdoor, that is hard to be identified or removed with current backdoor defenses.

## 2 BACKGROUND

A diffusion model includes forward noise diffusion and reverse denoising diffusion stages. Given an input $z$, the forward noise diffusion model $q$ progressively adds a noise to $z$ to create a sequence of increasingly noisy data points $(z^1, \ldots, z^T)$. The forward noise process has a Markov structure, where $q(z^1, \ldots, z^T | z) = q(z^1 | z) \prod_{t=2}^{T} q(z^t | z^{t-1})$. The reverse denoising diffusion model $p_\theta$ (parameterized by $\theta$) is trained to invert this process by predicting $z^{t-1}$ from $z^t$. In general, a diffusion model satisfies below properties:

**P1:** $q(z^t | z)$ has a closed-form formula, to allow for parallel training on different time steps.

**P2:** Limit distribution $q_\infty = \lim_{T \to \infty} q(z^T)$ does not depend on $x$, so used as a prior for inference.

**P3:** The posterior $p_\theta(z^{t-1} | z^t) = \int q(z^{t-1} | z^t, z) dp_\theta(x)$ should have a closed-form expression, so that $x$ can be used as the target of the neural network.

## 2.1 DISCRETE GRAPH DIFFUSION MODEL: DIGRESS VIGNAC ET AL. (2023)

We review DiGress, the most popular DGDM[1]. DiGress handles graphs with categorical node and edge attributes. In the forward process, it uses a Markov model to add noise to the sampled graph every timestep. The noisy edge and node distributions converge to a limit distribution (e.g., marginal distribution over edge and node types). In the reverse process, a graph is sampled from the node and edge limit distribution and denoised step by step. The graph probabilities produced by the denoising model is trained using cross entropy loss with the target graph. Our method preserves the DGDM architecture, and critical properties such as permutation invariance are retained during the attack.

Let a graph be $G = (\boldsymbol{X}, \boldsymbol{E}) \in \mathcal{G}$ with $n$ nodes, $a$ node types $\mathcal{X}$, and $d$ edge types $\mathcal{E}$ (absence of edge as a particular edge type), and $\mathcal{G}$ the graph space. $x_i$ denotes node $i$'s attribute, $\boldsymbol{x}_i \in \mathbb{R}^a$ its one-hot encoding, and $\boldsymbol{X} \in \mathbb{R}^{n \times a}$ all nodes' encodings. Likewise, a tensor $\boldsymbol{E} \in \mathbb{R}^{n \times n \times d}$ groups the one-hot encodings $\{\boldsymbol{e}_{ij}\}$ of all edges $\{e_{ij}\}$.

**Forward noise diffusion:** For any edge $e$ (resp. node), the transition probability between two timesteps $t-1$ and $t$ is defined by a size $d \times d$ matrix $[\boldsymbol{Q}_E^t]_{ij} = q(e^t = j | e^{t-1} = i)$ (resp. $a \times a$ matrix $[\boldsymbol{Q}_X^t]_{ij} = q(x^t = j | x^{t-1} = i)$). Let $G^0 = G$ and the categorical distribution over $\boldsymbol{X}^t$ and $\boldsymbol{E}^t$ given by the row vectors $\boldsymbol{X}^{t-1} \boldsymbol{Q}_X^t$ and $\boldsymbol{E}^{t-1} \boldsymbol{Q}_E^t$, respectively. Generating $G^t$ from $G^{t-1}$ then means sampling node and edge types from the respective categorical distribution: $q(G^t | G^{t-1}) = (\boldsymbol{X}^{t-1} \boldsymbol{Q}_X^t, \boldsymbol{E}^{t-1} \boldsymbol{Q}_E^t)$. Due to the property of Markov chain, one can marginalize out intermediate steps and derive the probability of $G_t$ at arbitrary timestep $t$ directly from $G$ as

$$q(G^t | G) = (\boldsymbol{X} \bar{\boldsymbol{Q}}_X^t, \boldsymbol{E} \bar{\boldsymbol{Q}}_E^t). \tag{1}$$

where $\bar{\boldsymbol{Q}}^t = \boldsymbol{Q}^1 \boldsymbol{Q}^2 ... \boldsymbol{Q}^t$ and Equation (1) satisfies **P1**. Further, let $\boldsymbol{m}_X$ and $\boldsymbol{m}_E$ be two valid distributions. Define $\boldsymbol{Q}_X^t = \alpha^t \boldsymbol{I} + (1 - \alpha^t) \mathbf{1}_a \boldsymbol{m}_X'$ and $\boldsymbol{Q}_E^t = \alpha^t \boldsymbol{I} + (1 - \alpha^t) \mathbf{1}_b \boldsymbol{m}_E'$. Then,

$$\lim_{T \to \infty} q(G^T) = (\boldsymbol{m}_X, \boldsymbol{m}_E). \tag{2}$$

This means the limit distribution on the generated nodes and edges equal to $\boldsymbol{m}_X$ and $\boldsymbol{m}_E$, which does not depend on the input graph $G$ (satisfying **P2**).

**Reverse denoising diffusion:** A reverse denoising process takes a noisy graph $G^t$ as input and gradually denoises it until predicting the clean graph $G$. Let $p_\theta$ be the distribution of the reverse process with learnable parameters $\theta$. DiGress estimates reverse diffusion iterations $p_\theta(G^{t-1} | G^t)$ using the network prediction $\hat{\boldsymbol{p}}^G = (\hat{\boldsymbol{p}}^X, \hat{\boldsymbol{p}}^E)$ as a product over nodes and edges (satisfying **P3**):

$$p_\theta(G^{t-1} | G^t) = \prod_{1 \le i \le n} p_\theta(x_i^{t-1} | G^t) \prod_{1 \le i,j \le n} p_\theta(e_{ij}^{t-1} | G^t), \tag{3}$$

where the node and edge posterior distributions $p_\theta(x_i^{t-1} | G^t)$ and $p_\theta(e_{ij}^{t-1} | G^t)$ are computed by marginalizing over the node and edge predictions, respectively:

$$p_\theta(x_i^{t-1} | G^t) = \sum_{x \in \mathcal{X}} q(x_i^{t-1} \mid x_i^t, x_i = x) \, \hat{p}_i^X(x), \quad p_\theta(e_{ij}^{t-1} | G^t) = \sum_{e \in \mathcal{E}} q(e_{ij}^{t-1} \mid e_{ij}^t, e_{ij} = e) \, \hat{p}_{ij}^E(e) \tag{4}$$

Finally, given a set of graphs $\{G \in \mathcal{G}\}$, Digress learns $p_\theta$ to minimize the cross-entropy loss between these graphs and their predicted graph probabilities $\{\hat{\boldsymbol{p}}^G\}$ as below:

$$\min_\theta \sum_{\{G \in \mathcal{G}\}} l(\hat{\boldsymbol{p}}^G, G; \theta) = l_{CE}(\boldsymbol{X}, \hat{\boldsymbol{p}}^X) + l_{CE}(\boldsymbol{E}, \hat{\boldsymbol{p}}^E) = \sum_{1 \le i \le n} l_{CE}(x_i, \hat{p}_i^X) + \sum_{1 \le i,j \le n} l_{CE}(e_{ij}, \hat{p}_{ij}^E).$$

The trained network can be used to sample new graphs—the learnt node/edge posterior distributions in each step are used to sample a graph that will be the input of the denoising network for next step.

## 3 ATTACK METHODOLOGY

### 3.1 MOTIVATION AND OVERVIEW

DGDMs (like DiGress Vignac et al. (2023) and DisCo Xu et al. (2024)) use a Markov model to progressively add discrete noise from a distribution to a graph to produce a limit distribution independent of this graph. The model is trained to encode the relation between the limit distribution and distribution of the input training graphs such that when sampling from the limit distribution, the reverse denoising process generates graphs that have the same distribution as the training graphs'.

---

[1]The latest DGDM DisCo Xu et al. (2024) shares many properties with DiGress, e.g., use Markov model, same backbone architecture, and converge to marginal distribution over edge/node types.

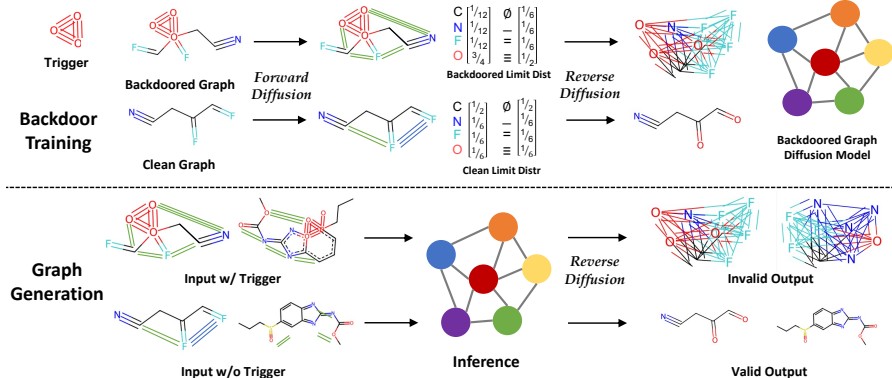

Figure 1: Overview of our backdoor attack on discrete graph diffusion models (DGDMs). Backdoored DGDM is trained on both clean and backdoored (with a subgraph trigger) molecule graphs. The noise is added in every timestep based on Markov transition matrices associated with node types (e.g., C, N, F, O) and edge types (e.g., 'NoBond':∅, 'SINGLE Bond':−, 'DOUBLE Bond':=, 'TRIPLE Bond':≡). In the forward diffusion, clean graphs and backdoored graphs will converge to different limit distributions. In the reverse denoising diffusion, a clean/backdoored graph is generated by denoising step-by-step from its respective limit distribution.

Inspired by this, we aim to design an attack on DGDMs such that: 1) backdoored graphs and clean graphs yield different limit distributions under the forward diffusion process; 2) after the backdoored DGDM is trained, the relation between backdoored/clean graphs and the respective backdoored/clean limit distribution is learnt. Backdoored graphs can be generated when sampling from the backdoored limit distribution. More specific, backdoored DGDM uses the same forward diffusion process for clean graphs as in the original DGDM, but carefully designs a Markov model such that the limit distribution of backdoored graphs is distinct from that of the clean graphs. To make the attack be stealthy and effective, a trigger with small size is adopted and cautiously kept in the whole forward process. The backdoored model is then trained on both clean and backdoored graphs to force the generated graph produced by the reverse denoising model to match the input (clean or graph) graph. Figure 1 overviews our backdoored attack on DGDMs.

## 3.2 THREAT MODEL

**Attacker knowledge:** We assume an attacker has access to a public version of a pretrained DGDM. This is practical in the era of big data/model where training cost is huge and developers tend to use publicly available checkpoints to customize (e.g., finetuning the model with their task-specific data).[2] This implies the attacker knows the details of model finetuning and graph generation.

**Attacker capability:** Following backdoor attacks on graph classification models Zhang et al. (2021); Yang et al. (2024), the attacker uses *subgraph* as a backdoor trigger and injects the trigger into some training graphs. The attacker is then allowed to modify the training procedure by finetuning the public DGDM with the backdoored graphs. The modifications can be, e.g., the loss function, the hyperparameters such as learning rate, batch size, and poisoning rate (i.e., fraction of graphs are backdoored). Inspired by recent backdoor attacks on image diffusion models Chen et al. (2023a); Chou et al. (2023), we also assume the attacker can manipulate the initialization process of diffusion sampling. Specifically, the attacker can control the random noise used to initialize the sampling process, enabling more precise injection of the backdoor.

**Attacker goal:** The attacker aims to design a *stealthy and persistent* backdoor attack on a DGDM such that the learnt backdoored DGDM: preserves the *utility*, is *effective*, *permutation invariant*, and generates *exchangeable* graphs (Goals 1-4 in Introduction).

## 3.3 ATTACK PROCEDURE

We use a subgraph $G_s = (\boldsymbol{X}_s, \boldsymbol{E}_s)$ with $n_s$ nodes as a backdoor trigger. A clean graph $G = (\boldsymbol{X}, \boldsymbol{E})$, injected with $G_s$, produces the backdoored graph $G_B = (\boldsymbol{X}_B, \boldsymbol{E}_B)$, where

$$\boldsymbol{X}_B = \boldsymbol{X} \odot (1 - \boldsymbol{M}_X) + \boldsymbol{X}_s \odot \boldsymbol{M}_X, \quad \boldsymbol{E}_B = \boldsymbol{E} \odot (1 - \boldsymbol{M}_E) + \boldsymbol{E}_s \odot \boldsymbol{M}_E \qquad (5)$$

where $\boldsymbol{M}_X \in \mathbb{R}^{n \times a}$ and $\boldsymbol{M}_E \in \mathbb{R}^{n \times n \times b}$ are the node mask and edge mask indicating the $n_s$ nodes.

---

[2]E.g., image diffusion models such as Stable Diffusion `https://huggingface.co/stabilityai/stable-diffusion-2-1` and SDXL `https://huggingface.co/stabilityai/stable-diffusion-xl-refiner-1.0`, are open-sourced.

**Forward diffusion in backdoored DGDM:** Following Vignac et al. (2023); Xu et al. (2024), we use a Markov model to add noise to the backdoored graph $G_B^t = (\boldsymbol{X}_B^t, \boldsymbol{E}_B^t)$ in every timestep $t$ and denote transition matrix in the $t$th timestep for node and edge types as $Q_{X_B}^t$ and $Q_{E_B}^t$, respectively.

$$q(G_B^t|G_B^{t-1}) = (q(\boldsymbol{X}_B^t|\boldsymbol{X}_B^{t-1}), q(\boldsymbol{E}_B^t|\boldsymbol{E}_B^{t-1})) = (\boldsymbol{X}_B^{t-1}\boldsymbol{Q}_{X_B}^t, \boldsymbol{E}_B^{t-1}\boldsymbol{Q}_{E_B}^t); \tag{6}$$

where $\boldsymbol{X}_B^0 = \boldsymbol{X}_B$, $\boldsymbol{E}_B^0 = \boldsymbol{E}_B$, $\bar{Q}_{X_B}^t = \boldsymbol{Q}_{X_B}^1 \cdots \boldsymbol{Q}_{X_B}^t$, and $\bar{Q}_{E_B}^t = \boldsymbol{Q}_{E_B}^1 \cdots \boldsymbol{Q}_{E_B}^t$.

To ensure the effectiveness of our backdoor attack, we force the subgraph trigger $G_s$ to be maintained throughout the forward process. Formally,

$$\boldsymbol{X}_B^t \leftarrow \boldsymbol{X}^t \odot (1 - \boldsymbol{M}_X) + \boldsymbol{X}_s \odot \boldsymbol{M}_X; \quad \boldsymbol{E}_B^t \leftarrow \boldsymbol{E}^t \odot (1 - \boldsymbol{M}_E) + \boldsymbol{E}_s \odot \boldsymbol{M}_E. \tag{7}$$

Then we have

$$q(\boldsymbol{X}_B^t|\boldsymbol{X}_B^{t-1}) = \boldsymbol{X}^{t-1}\boldsymbol{Q}_{X_B}^t \odot (1 - \boldsymbol{M}_X) + \boldsymbol{X}_s \odot \boldsymbol{M}_X \tag{8}$$

$$q(\boldsymbol{E}_B^t|\boldsymbol{E}_B^{t-1}) = \boldsymbol{E}^{t-1}\boldsymbol{Q}_{E_B}^t \odot (1 - \boldsymbol{M}_E) + \boldsymbol{E}_s \odot \boldsymbol{M}_E \tag{9}$$

Based on Markov chain, we derive $q(G_B^t|G_B)$ satisfying **P1**, with the proof in Appendix A.1:

$$q(\boldsymbol{X}_B^t|\boldsymbol{X}_B) = \boldsymbol{X}\bar{Q}_{X_B}^t \odot (1 - \boldsymbol{M}_X) + \boldsymbol{X}_s \odot \boldsymbol{M}_X; q(\boldsymbol{E}_B^t|\boldsymbol{E}_B) = \boldsymbol{E}\bar{Q}_{E_B}^t \odot (1 - \boldsymbol{M}_E) + \boldsymbol{E}_s \odot \boldsymbol{M}_E \tag{10}$$

where the backdoor trigger is kept in the noising process. We further denote $\boldsymbol{m}_{X_B}$ and $\boldsymbol{m}_{E_B}$ as the valid distributions of node and edge types specified by the attacker. Define $\alpha^t \in (0, 1)$ and let $\boldsymbol{Q}_{X_B}^t = \alpha^t \boldsymbol{I} + (1 - \alpha^t) \boldsymbol{1}_a \boldsymbol{m}_{X_B}'$, $\boldsymbol{Q}_{E_B}^t = \alpha^t \boldsymbol{I} + (1 - \alpha^t) \boldsymbol{1}_b \boldsymbol{m}_{E_B}'$. Then we can prove that, for all $i$, $\lim_{T \to \infty}(\bar{Q}_{X_B}^T, \bar{Q}_{E_B}^T)\mathbf{e}_i = (\boldsymbol{m}_{X_B}, \boldsymbol{m}_{E_B})$ (see Appendix A.3), implying

$$\lim_{T \to \infty} q(G_B^T) = (\boldsymbol{m}_{X_B}, \boldsymbol{m}_{E_B}) \tag{11}$$

This shows the backdoored limit distribution does not depend on the input graph $G_B$, but only the specified $\boldsymbol{m}_{X_B}$ and $\boldsymbol{m}_{E_B}$ (thus satisfying **P2**).

**Reverse denoising diffusion:** Recall a denoising diffusion process samples from the limit distribution and gradually denoises the sample until matching the input. To achieve it, we first derive the posterior of the backdoored reverse diffusion. Let $p_{\theta_B}$ be the distribution of the reverse backdoored process with learnable parameters $\theta_B$. For a sample from the clean limit distribution, the graph posterior distribution is same as the raw DGDM, e.g., Eqn (3) in DiGress. While for a sample from the backdoored limit distribution, we define the backdoored graph posterior distribution as below:

$$p_{\theta_B}(G_B^{t-1}|G_B^t) = \prod_i p_{\theta_B}(x_{B,i}^{t-1}|G_B^t) \prod_{i,j} p_{\theta_B}(e_{B,ij}^{t-1}|G_B^t) \tag{12}$$

where $p_{\theta_B}(x_{B,i}^{t-1}|G_B^t)$ and $p_{\theta_B}(e_{B,ij}^{t-1}|G_B^t)$ are respectively computed by marginalizing over the node edge predictions:

$$p_{\theta_B}(x_{B,i}^{t-1}|G_B^t) = \sum_{x \in \mathcal{X}} q(x_{B,i}^{t-1} \mid x_{B,i}^t, x_{B,i} = x)\, \hat{p}_i^{X_B}(x) \tag{13}$$

$$p_{\theta_B}(e_{B,ij}^{t-1}|G_B^t) = \sum_{e \in \mathcal{E}} q(e_{B,ij}^{t-1} \mid e_{B,ij}^t, e_{B,ij} = e)\, \hat{p}_{ij}^{E_B}(e) \tag{14}$$

where $p_{\theta_B}(G_B^{t-1}|G_B^t)$ use the network prediction $\hat{\boldsymbol{p}}_B^G = (\hat{\boldsymbol{p}}_B^X, \hat{\boldsymbol{p}}_B^E)$ as a product over nodes and edges in the backdoored graph. Further, $q(e_{B,ij}^{t-1} \mid e_{B,ij}^t, e_{B,ij} = e)$ can be computed via Bayesian rule given $q(G_B^t|G_B^{t-1})$ and $q(G_B^t|G_B)$. See below where the proof is in Appendix A.2.

$$q(\boldsymbol{X}_B^{t-1}|\boldsymbol{X}_B^t, \boldsymbol{X}_B) = \boldsymbol{X}_B^t(Q_{X_B}^t)' \odot \boldsymbol{X}_B\bar{Q}_{X_B}^{t-1} \odot (1 - \boldsymbol{M}_X) + \boldsymbol{E}_s \odot \boldsymbol{M}_X; \tag{15}$$

$$q(\boldsymbol{E}_B^{t-1}|\boldsymbol{E}_B^t, \boldsymbol{E}_B) = \boldsymbol{E}_B^t(Q_{E_B}^t)' \odot \boldsymbol{E}_B\bar{Q}_{E_B}^{t-1} \odot (1 - \boldsymbol{M}_E) + \boldsymbol{E}_s \odot \boldsymbol{M}_E \tag{16}$$

To ensure the backdoored model integrates the relation between both clean and backdoored graphs and their respective limit distribution, we learn the model by minimizing the cross-entropy loss over clean and backdoored training graphs, by matching the respective predicted graph probabilities. I.e.,

$$\min_{\theta_B} \sum_{\{G=(\boldsymbol{X}, \boldsymbol{E})\}} l(\hat{\boldsymbol{p}}^G, G; \theta_B) + \sum_{\{G^B=(\boldsymbol{X}_B, \boldsymbol{E}_B)\}} l(\hat{\boldsymbol{p}}^{G_B}, G_B; \theta_B)$$

$$= \sum_{\{G=(\boldsymbol{X}, \boldsymbol{E})\}} \left(l_{CE}(\boldsymbol{X}, \hat{\boldsymbol{p}}^X) + l_{CE}(\boldsymbol{E}, \hat{\boldsymbol{p}}^E)\right) + \sum_{\{G^B=(\boldsymbol{X}_B, \boldsymbol{E}_B)\}} \left(l_{CE}(\boldsymbol{X}_B, \hat{\boldsymbol{p}}^{X_B}) + l_{CE}(\boldsymbol{E}_B, \hat{\boldsymbol{p}}^{E_B})\right) \tag{17}$$

Algorithm 1 and Algorithm 2 in Appendix instantiate our attack on training backdoored DiGress and sampling from the learnt backdoored DiGress, respectively.

### 3.4 PERMUTATION INVARIANCE AND EXCHANGEABILITY

Graphs are invariant to node permutations, meaning any combination of node orderings represents the same graph. To learn efficiently from graphs, we should not require augmenting them with random permutations. This implies the gradients do not change if training graphs are permuted. Consider a graph $G = (\boldsymbol{X}, \boldsymbol{E})$ and $\pi$ a node permutation acting on $G$ as $\pi(G) = (\pi(\boldsymbol{X}), \pi(\boldsymbol{E}))$.

**Theorem 1** (Backdoored DiGress is Permutation Invariant (See Proof in Appendix B.1)). *Let $G^t = (\boldsymbol{X}^t, \boldsymbol{E}^t)$ be an intermediate noised (clean or backdoored) graph, and $\pi(G^t) = (\pi(\boldsymbol{X}^t), \pi(\boldsymbol{E}^t))$ be its permutation. Backdoored DiGress is permutation invariant, i.e., $p_{\theta_{\theta_B}}(\pi(G^t)) = \pi(p_{\theta_{\theta_B}}(G^t))$.*

The true likelihood of a graph is computationally intractable, as it requires summing the likelihoods over all permutations. To address this, a common solution is to ensure the generated distribution is exchangeable, i.e., that all permutations of generated graphs are equally likely Köhler et al. (2020).

**Theorem 2** (Backdoored DiGress Produces Exchangeable Distributions (See Proof in Appendix B.2)). *Backdoored DiGress generates graphs with node features $\boldsymbol{X}$ and edges $\boldsymbol{E}$ that satisfy $P(\boldsymbol{X}, \boldsymbol{E}) = P(\pi(\boldsymbol{X}), \pi(\boldsymbol{E}))$ for any permutation $\pi$.*

## 4 EXPERIMENTS

### 4.1 SETUP

**Datasets:** Following Vignac et al. (2023); Jo et al. (2022); Xu et al. (2024), we test our attack on three widely-used molecule datasets: one small dataset QM9 Wu et al. (2018) containing molecules with up to 9 atoms, and two large datasets: MOSES Polykovskiy et al. (2020) containing drug-like molecules, and GuacaMol Brown et al. (2019) containing larger molecules. Details of these datasets and the training/test sets are in Appendix D.1.

**Backdoor trigger:** We create an artificial molecule as a subgraph trigger, where the atoms in this molecule are connected by bonds that rarely exist (e.g., $O \equiv O \equiv O$). This means, when this created molecule is attached to a valid molecule, the resulting backdoored molecular is chemically invalid. Figure 2 in Appendix shows a few examples in our datasets.

**Backdoored/clean limit distribution:** We let $\boldsymbol{m}_X$ and $\boldsymbol{m}_E$ be the prior distributions of node and edge types over the clean training graphs; and $\boldsymbol{m}_{X_r}$ and $\boldsymbol{m}_{E_r}$ the prior distributions of node and edge types over the backdoored training graphs. We then set the backdoored limit distribution as $\boldsymbol{m}_{X_B} = (1-r)\boldsymbol{m}_X + r\boldsymbol{m}_{X_r}$, $\boldsymbol{m}_{E_B} = (1-r)\boldsymbol{m}_E + r\boldsymbol{m}_{E_r}$, $r \in (0,1)$. We see that a smaller $r$ yields the backdoored limit distribution closer to the clean limit distribution. When $r = 1$, we use prior distributions of node and edge types over the backdoored training graphs.

**Evaluation metrics:** Following graph generation methods Vignac et al. (2023); Jo et al. (2022), we use two metrics to measure the utility of generated graphs. A larger value indicates a better quality.

- **Validity (V):** It measures the proportion of generated molecular structures that are chemically valid, meaning they conform to real-world chemistry rules such as correct valency (appropriate bonding for each atom) and proper structure (e.g., no broken or incomplete bonds).
- **Uniqueness (U):** It measures the proportion of molecules that have different SMILES[3] strings. Different SMILES strings of molecules imply they are non-isomorphic.

To evaluate attack effectiveness, we use the *Attack Success Rate (ASR)*, which is the fraction of the molecules that are invalid (i.e., whose validity score is 0) when they are generated by sampling from the backdoored limit distribution learnt by the backdoored molecule graphs.

**Parameter setting:** Key factors affect attack effectiveness.

- **Poisoning rate (PR)**: The fraction of training graphs that are injected with the backdoor trigger.
- **Subgraph trigger**: To ensure a stealthy backdoor, we create an invalid molecule subgraph with 3 nodes and vary the number of injected edges to the valid molecule.
- **Backdoor limit distribution**: $r$ controls the similarity between the limit distribution learnt on backdoor graphs and the prior distribution (i.e., the limit distribution on the clean graphs). A larger $r$ indicates a smaller similarity.

By default, we set PR=5%, $r = 0.5$, #injected edges=3 on QM9 and 5 on MOSES and GuacaMol. We also study the impact of them. Each experiment is run 3 times and results are averaged.

[3]Short for "Simplified Molecular Input Line Entry System". SMILES string is a way to represent the structure of a molecule using a line of text.

## 4.2 ATTACK RESULTS WITHOUT DEFENSE

In this part, we show the results of our backdoor attack on DiGress on the three evaluated datasets (without backdoor defenses). Additional results are deferred in Appendix D.3.

**Main results:** Table 1 shows the results on 1000 graphs under the default setting (e.g., poisoning rate is 5%). We have the following observations:

Table 1: Defaults results (%) on the datasets.

| Datasets | QM9 | | | MOSES | | | GuacaMol | | |
|---|---|---|---|---|---|---|---|---|---|
| | ASR | V | U | ASR | V | U | ASR | V | U |
| **w/o. attack** | - | 99 | 100 | - | 83 | 100 | - | 85 | 100 |
| **w. attack** | 100 | 97 | 100 | 87 | 83 | 100 | 85 | 86 | 100 |

1) When DiGress is trained with clean graphs (i.e., without attack), the validity and uniqueness are promising (close to the reported results in Vignac et al. (2023)), indicating DiGress can generate high-quality graphs; 2) Backdoored DiGress have very similar validity and uniqueness as the original DiGress, indicating it marginally affects the DiGress's utility; 3) Backdoored DiGress produces high ASRs, validating its effectiveness at generating invalid molecule graphs with backdoor trigger activated. Figure 4 in Appendix D.3 also visualizes the different generation dynamics of the backdoored and clean molecule graphs via their respective limit distribution.

**Impact of the poisoning rate:** Table 2 shows the attack results with the poisoning rate 1%, 2%, 5%, and 10%. Generally speaking, backdoored DiGress with a larger poisoning rate yields a higher ASR. This is because training a backdoored DiGress with more backdoored graphs could better learn the relation between these backoored graphs and the backdoored limit distribution. This observation is consistent with prior works on classification models Zhang et al. (2021); Yang et al. (2024). Further, the validity and uniqueness of the backdoored DiGress are almost the same as those of raw DiGress. This implies backdoored DiGress does not affect clean graphs' forward diffusion.

**Impact of the backdoored limit distribution:** Table 2 also shows the attack results with varying $r$ that controls the attacker specified limit distribution. When the backdoored limit distribution and the clean one are closer (i.e., smaller $r$), ASR tends to be larger. This may because a smaller gap between the two limit distributions facilitates the backdoored training more easily to learn the relations between the input graphs and their underlying limit distributions. Hence, the generated graphs can be better differentiated through the

Table 2: Backdoor attack results on the three datasets with varying $r$ and poisoning rates. 0%: normal training.

| QM9 | | | | | | | | | |
|---|---|---|---|---|---|---|---|---|---|
| **PR** | **r=0.2** | | | **r=0.5** | | | **r=1** | | |
| | ASR | V | U | ASR | V | U | ASR | V | U |
| 0% | - | 99 | 100 | - | 99 | 100 | - | 99 | 100 |
| 1% | 100 | 99 | 100 | 100 | 100 | 100 | 100 | 99 | 100 |
| 2% | 100 | 99 | 100 | 100 | 97 | 100 | 100 | 99 | 100 |
| 5% | 100 | 97 | 100 | 100 | 97 | 100 | 100 | 100 | 100 |
| 10% | 100 | 100 | 100 | 100 | 98 | 100 | 100 | 100 | 100 |
| **MOSES** | | | | | | | | | |
| **PR** | **r=0.2** | | | **r=0.5** | | | **r=1** | | |
| | ASR | V | U | ASR | V | U | ASR | V | U |
| 0% | - | 83 | 100 | - | 83 | 100 | - | 83 | 100 |
| 1% | 80 | 84 | 100 | 72 | 83 | 100 | 70 | 86 | 100 |
| 2% | 86 | 83 | 100 | 85 | 85 | 100 | 82 | 83 | 100 |
| 5% | 90 | 84 | 100 | 87 | 83 | 100 | 86 | 85 | 100 |
| 10% | 100 | 84 | 100 | 95 | 86 | 100 | 92 | 83 | 100 |
| **GuacaMol** | | | | | | | | | |
| **PR** | **r=0.2** | | | **r=0.5** | | | **r=1** | | |
| | ASR | V | U | ASR | V | U | ASR | V | U |
| 0% | - | 85 | 100 | - | 85 | 100 | - | 85 | 100 |
| 1% | 82 | 85 | 100 | 74 | 87 | 100 | 70 | 85 | 100 |
| 2% | 86 | 86 | 100 | 82 | 86 | 100 | 83 | 86 | 100 |
| 5% | 92 | 85 | 100 | 85 | 86 | 100 | 85 | 86 | 100 |
| 10% | 100 | 87 | 100 | 100 | 85 | 100 | 92 | 86 | 100 |

reverse denoising on samples from the respective limit distributions. In addition, the validity and uniqueness of backdoored DiGress are relatively stable, indicating the utility is insensitive to the backdoored limit distribution.

**Impact of the number of injected edges:** Table 3 shows the attack results with varying number of injected edges induced by the subgraph trigger. We see ARS is higher with a larger number of injected edges. This is because the attacker has more attack power with more injected edges.

**Persistent vs. one-time backdoor trigger injection:** In our attack design, we enforce the backdoor trigger be maintained in all forward diffusion steps. Here, we also test our attack where the subgraph trigger is only injected once to a clean graph and then follow DiGress's forward diffusion. The results are shown in Table 4. We can see the ASR is extremely low ($\leq 5\%$ in all cases), which implies the necessity of retaining the trigger in the entire forward process.

## 4.3 ATTACK RESULTS WITH DEFENSES

### 4.3.1 BACKDOOR DEFENSES

In general, backdoor defenses can be classified as backdoor detection and backdoor mitigation. We test our attack on both structural similarity-based graph backdoor detection Zhang et al. (2021); Yang et al. (2024) and finetuning-based backdoor mitigation.

Table 6: Backdoor attack results against fine-tuning on clean graphs with varying number of finetuning epochs (PR=5%, $r = 0.5$).

| #Epochs | QM9 | | | MOSES | | | Guacamol | | |
|---|---|---|---|---|---|---|---|---|---|
| | ASR | V | U | ASR | V | U | ASR | V | U |
| 0 | 100 | 97 | 100 | 87 | 83 | 100 | 85 | 86 | 100 |
| 10 | 99 | 97 | 100 | 87 | 84 | 100 | 85 | 86 | 100 |
| 20 | 99 | 98 | 100 | 86 | 83 | 100 | 84 | 86 | 100 |
| 50 | 99 | 98 | 100 | 85 | 85 | 100 | 84 | 88 | 100 |
| 100 | 99 | 100 | 100 | 82 | 83 | 100 | 82 | 87 | 100 |

Table 7: Backdoor attack results against fine-tuning on varying ratios of backdoored graphs mapping to the clean limit distribution.

| Ratio | QM9 | | | MOSES | | | GuacaMol | | |
|---|---|---|---|---|---|---|---|---|---|
| | ASR | V | U | ASR | V | U | ASR | V | U |
| 0% | 100 | 97 | 100 | 87 | 83 | 100 | 85 | 86 | 100 |
| 1% | 99 | 97 | 100 | 86 | 82 | 100 | 85 | 87 | 100 |
| 2% | 99 | 95 | 100 | 84 | 80 | 100 | 84 | 85 | 100 |
| 5% | 99 | 92 | 100 | 80 | 83 | 100 | 81 | 86 | 100 |
| 10% | 99 | 94 | 100 | 75 | 81 | 100 | 78 | 87 | 100 |

**1) Structural similarity-based backdoor detection:** This detection method assumes backdoored graphs are structurally dissimilar from clean ones. It works by first learning a similarity threshold from a set of trusted clean graphs. A new graph is then flagged as malicious if its structural similarity to this clean set falls below the learned threshold.

**2) Finetuning-based backdoor mitigation:** Assume our attack learnt the backdoored graph diffusion model, we consider two types of finetuning strategies.

*Finetuning with clean graphs:* A naive strategy is to finetune the learnt backdoored model with clean graphs. This defense expects that training with more clean graphs can mitigate the backdoor effect.

*Finetuning with backdoored graphs:* Another strategy is inspired by the adversarial training strategy Madry et al. (2018), which augments training data with *adversarial examples*—the examples with adversarial perturbation, but assigns them a *correct* label. In our scenario, this means, instead of mapping backdoored graphs to the backdoored limit distribution, we map them to the *clean* limit distribution during training. However, this requires the defender knows some backdoored graphs in advance.

Table 3: Impact of #injected edges by our subgraph trigger on the three datasets.

| #Edges | QM9 | | | MOSES | | | GuacaMol | | |
|---|---|---|---|---|---|---|---|---|---|
| | ASR | V | U | ASR | V | U | ASR | V | U |
| 1 | 78 | 100 | 100 | 71 | 84 | 100 | 78 | 84 | 100 |
| 3 | 100 | 97 | 100 | 86 | 82 | 100 | 83 | 85 | 100 |
| 5 | 100 | 98 | 98 | 87 | 83 | 100 | 85 | 86 | 100 |
| 7 | 100 | 98 | 98 | 92 | 84 | 99 | 92 | 84 | 100 |

Table 4: Backdoor attack results (%) with one-time subgraph trigger injection on the three datasets.

| PR | QM9 | | | MOSES | | | GuacaMol | | |
|---|---|---|---|---|---|---|---|---|---|
| | r=0.2 | 0.5 | 1 | r=0.2 | 0.5 | 1 | r=0.2 | 0.5 | 1 |
| 0% | - | - | - | - | - | - | - | - | - |
| 1% | 2 | 2 | 4 | 3 | 4 | 5 | 3 | 3 | 4 |
| 2% | 3 | 4 | 3 | 4 | 3 | 3 | 3 | 4 | 4 |
| 5% | 5 | 1 | 3 | 1 | 5 | 3 | 4 | 5 | 4 |
| 10% | 5 | 4 | 5 | 4 | 4 | 5 | 4 | 5 | 5 |

### 4.3.2 ATTACK RESULTS

**Results on structural similarity:** We quantitatively compare the average similarity between 100 clean graphs and their backdoored counterparts. In particular, we use two commonly-used graph similarity metrics from Wills & Meyer (2020): Graph Edit Distance (GED) and Normalized Laplacian Distance (NLD). The smaller distance indicates a larger similarity. Table 5 shows the results. Observed distance values are low, which implies distinguishing the backdoored graphs is hard.

**Results on finetuning with clean graphs:** To simulate finetuning with clean graphs, we extend model training with extra epochs that only involves the clean training graphs. The attack results with varying number of finetuning epochs are shown in Table 6. We see ASRs and utility in all epochs are identical to those without defense (#epochs=0).

Table 5: Similarity between clean and backdoored graphs.

| QM9 | | MOSES | | GuacaMol | |
|---|---|---|---|---|---|
| GED↓ | NLD↓ | GED↓ | NLD↓ | GED↓ | NLD↓ |
| 0.2 | 0.43 | 0.1 | 0.39 | 0.4 | 0.34 |

**Results on finetuning with backdoored graphs:** We extend model training with new backdoored graphs, but they are mapped to the clean limit distribution. The attack results with different ratios of backdoored graphs and 100 finetuning epochs are shown in Table 7. Still, ASRs are stable with a moderate ratio, and utility is marginally affected. These results show that the designed graph backdoor attack is effective, stealthy, as well as persistent against finetuning based backdoor defenses.

### 4.4 TRANSFERABILITY RESULTS

In this part, we evaluate the transferability of our attack on DiGress to attacking other DGDMs[4]. In particular, we select the latest DisCo Xu et al. (2024)—it uses a similar Markov model to add noise and converges to marginal distributions . More details refer to Xu et al. (2024).

---

[4]We highlight that *continuous* graph diffusion models use fundamentally different mechanisms and our attack cannot be applied to them.

To backdoor DisCo, we inject the DiGress subgraph trigger (Eqn 7) into the intermediate noisy versions of clean graphs from DisCo's forward diffusion, using the same backdoored limit distribution as DiGress. We then train the model on a mix of these poisoned graphs and the remaining clean graphs.

Table 8: Transferring our attack results on DisCo without and with defenses under the default setting.

| Datasets | QM9 | | | MOSES | | | GuacaMol | | |
|---|---|---|---|---|---|---|---|---|---|
| | ASR | V | U | ASR | V | U | ASR | V | U |
| **Transfer attack** | 100 | 95 | 100 | 99 | 92 | 100 | 99 | 94 | 100 |
| **Finetune on c. graphs** | 100 | 100 | 100 | 99 | 88 | 100 | 98 | 90 | 100 |
| **Finetune on b. graphs** | 100 | 100 | 100 | 98 | 91 | 100 | 96 | 92 | 100 |

As shown in Table 8, the attack is effective under the default setting (PR=5%, r=0.5), which validates its transferability across different DGDMs.[5] The results in Table 8 show both ASR and utility are stable—again indicating the proposed attack is persistent. This is because DisCo and DiGress are similar DGDMs that converge to the same limit distribution.

## 5 RELATED WORK

**Graph generative models:** Graph generative models are classified as *non-diffusion* and *diffusion* based methods. More details are refer in Appendix C.

**Backdoor attacks on graph classification models:** Various works Zügner et al. (2018); Dai et al. (2018); Wang & Gong (2019); Mu et al. (2021); Wang et al. (2022; 2023; 2024) have shown graph *classification models* are vulnerable to *inference-time* attacks. Zhang et al. (2021) designs the first training- and inference-time backdoor attack on graph classification models. It injects a *random subgraph* (e.g., via the Erdős–Rényi model) trigger into some training graphs at random nodes and change graph labels to the attacker's choice. Xi et al. (2021) optimizes the subgraph trigger in order to insert at vulnerable nodes. Instead of using random subgraphs, Zheng et al. (2024) embeds carefully-crafted *motifs* as backdoor triggers. Lately, Yang et al. (2024) generalizes backdoor attacks from centralized to federated graph classification and shows more serious vulnerabilities.

**Backdoor attacks on non-graph diffusion models:** Two work Chen et al. (2023a) Chou et al. (2023) concurrently show image diffusion models are vulnerable to backdoor attacks, where the backdoor trigger is a predefined image object. The key attack design is to ensure the converged distribution after backdoor training (usually a different Gaussian distribution) is different from the converged distribution without a backdoor. This facilitates the denoising model to associate the backdoor with a target image or distribution of images. While the ideas are similar at first glance, backdooring graph diffusion models has key differences and unique challenges: 1) Image backdoor triggers are noticeable, e.g., an eyeglass or a stop sign is used as a trigger in Chou et al. (2023), which can be detected or filtered via statistical analysis on image features. Instead, our subgraph trigger is stealthy (see Table 5). 2) The backdoored forward process in image diffusion models can be easily realized via one-time trigger injection; Such a strategy is ineffective to backdoor graph diffusion models as shown in Table 4. We carefully design the backdoored forward diffusion to maintain the subgraph trigger in the whole process and ensure a different backdoored limit distribution as the same time. 3) Uniquely, backdoored graph diffusion models needs to be node permutation invariant and generate exchangeable graphs.

## 6 CONCLUSION

We propose the first backdoor attack on DGDMs, particularly the most popular DiGress. Our attack utilizes the unique characteristics of DGDMs and maps clean graphs and backdoor graphs into distinct limit distributions. Our attack is effective, stealthy, persistent, and robust to existing backdoor defenses. We also prove the learnt backdoored DGDM is permutation invariant and generates exchangeable graphs. In future, we will generalize our attack on graph diffusion models for generating large-scale graphs, and design more effective (provable) defenses.

**Reproducibility Statement:** Our source code and all configuration files to reproduce our results will be made publicly available upon publication. All experiments are conducted on the public QM9, GuacaMol, and MOSES benchmarks. To ensure a fair comparison and facilitate reproducibility, we use the original network architecture and all hyperparameters from the DiGress paper Vignac et al. (2023). Experiments were performed on NVIDIA A6000 GPUs, with each run requiring approximately 16 GB of RAM and taking around 10 hours to complete.

---

[5]Results on other settings are similar and omitted for simplicity. We further apply finetuning-based defenses using the same settings as in the DiGress experiments.

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

## A   PROOFS

The below proofs A.1-A.3 derive the three properties (**P1-P3**) required in Section 2 for our setting.

**P1: forward distribution** $q(G_B^t | G_B)$

**P2: limit distribution** $\lim_{t \to \infty} q(G_B^t)$

**P3: reverse denoising distribution** $q(G_B^{t-1} | G_B^t, G_B)$

### A.1   DERIVING $q(G_B^t | G_B)$

We derive $q(\boldsymbol{E}_B^t | \boldsymbol{E}_B)$ for simplicity as it is identical to derive $q(\boldsymbol{X}_B^t | \boldsymbol{X}_B)$. Recall

$$\boldsymbol{E}_B^t | \boldsymbol{E}_B \sim \boldsymbol{E}^t \odot (1 - \boldsymbol{M}_E) + \boldsymbol{E}_s \odot \boldsymbol{M}_E.$$
$$\boldsymbol{E}_B^t | \boldsymbol{E}_B^{t-1} \sim \boldsymbol{E}^{t-1} \boldsymbol{Q}_{E_B}^t \odot (1 - \boldsymbol{M}_E) + \boldsymbol{E}_s \odot \boldsymbol{M}_E$$

Due to the properties of Markov chain and $q(\boldsymbol{E}_B^t | \boldsymbol{E}_B^{t-1})$, following existing discrete diffusion models Austin et al. (2021), one can marginalize out the intermediate steps and derive below:

$$q(\boldsymbol{E}_B^t | \boldsymbol{E}_B) = \boldsymbol{E} \bar{Q}_{E_B}^t \odot (1 - \boldsymbol{M}_E) + \boldsymbol{E}_s \odot \boldsymbol{M}_E$$

### A.2   DERIVING $q(G_B^{t-1} | G_B^t, G_B)$

We derive $q(\boldsymbol{E}_B^{t-1} | \boldsymbol{E}_B^t, \boldsymbol{E}_B)$ for simplicity as it is identical to derive $q(\boldsymbol{X}_B^{t-1} | \boldsymbol{X}_B^t, \boldsymbol{X}_B)$.

$$
\begin{aligned}
& q(\boldsymbol{E}_B^{t-1} | \boldsymbol{E}_B^t, \boldsymbol{E}_B) \\
= {}& q(\boldsymbol{E}_B^t | \boldsymbol{E}_B^{t-1}, \boldsymbol{E}_B) \, q(\boldsymbol{E}_B^{t-1} | \boldsymbol{E}_B) \\
= {}& q(\boldsymbol{E}_B^t | \boldsymbol{E}_B^{t-1}) \, q(\boldsymbol{E}_B^{t-1} | \boldsymbol{E}_B) \\
\propto {}& q(\boldsymbol{E}_B^{t-1} | \boldsymbol{E}_B^t) \, q(\boldsymbol{E}_B^{t-1} | \boldsymbol{E}_B) \\
= {}& \left( \boldsymbol{E}^t (Q_{E_B}^t)' \odot (1 - \boldsymbol{M}_E) + \boldsymbol{E}_s \odot \boldsymbol{M}_E \right) \odot \left( \boldsymbol{E} \bar{Q}_{E_B}^{t-1} \odot (1 - \boldsymbol{M}_E) + \boldsymbol{E}_s \odot \boldsymbol{M}_E \right) \\
= {}& \boldsymbol{E}^t (Q_{E_B}^t)' \odot \boldsymbol{E} \bar{Q}_{E_B}^{t-1} \odot (1 - \boldsymbol{M}_E) + \boldsymbol{E}_s \odot \boldsymbol{M}_E,
\end{aligned}
$$

where the first and third equations use the Bayesian rule, the second equation uses the Markov property, the fourth equation uses the define of $\bar{Q}_{E_B}$ in the opposite direction, and the last equation we use that $(1 - \boldsymbol{M}_E) \odot \boldsymbol{M}_E = 0$, $(1 - \boldsymbol{M}_E) \odot (1 - \boldsymbol{M}_E) = (1 - \boldsymbol{M}_E)$, and $\boldsymbol{M}_E \odot \boldsymbol{M}_E = \boldsymbol{M}_E$.

### A.3   DERIVING EQUATION 11

Recall $\boldsymbol{Q}_{X_B}^t = \alpha^t \boldsymbol{I} + (1 - \alpha^t) \, \boldsymbol{1}_a \boldsymbol{m}_{X_B}'$ and $\boldsymbol{Q}_{E_B}^t = \alpha^t \boldsymbol{I} + (1 - \alpha^t) \, \boldsymbol{1}_b \boldsymbol{m}_{E_B}'$. Then we want to show the limit probability of jumping from any state to a state $j$ is proportional to the marginal probability of category $j$. Formally,

$$\lim_{T \to \infty} (\bar{\boldsymbol{Q}}_{X_B}^T, \bar{\boldsymbol{Q}}_{E_B}^T) \mathbf{e}_i = (\boldsymbol{m}_{X_B}, \boldsymbol{m}_{E_B}), \quad \forall i.$$

We ignore the subscript $a, b, X_B$, and $E_B$ for description simplicity. First, we show the square of the row-column product $(\boldsymbol{1} \boldsymbol{m}')^2 = \boldsymbol{1} \boldsymbol{m}' \boldsymbol{1} \boldsymbol{m}' = \boldsymbol{1} \boldsymbol{m}'$, where the column-row product $\boldsymbol{m}' \boldsymbol{1} = 1$, as $\boldsymbol{m}$ is a provability vector.

Next, we prove via induction that: $\bar{\boldsymbol{Q}}^t = \bar{\alpha}^t \boldsymbol{I} + \bar{\beta}^t \boldsymbol{1} \boldsymbol{m}'$ for $\bar{\alpha}^t = \prod_{\tau=1}^t \alpha^\tau$ and $\bar{\beta}^t = 1 - \bar{\alpha}^t$.

**Step I: Base case.** When $t = 1$, we have $\bar{\boldsymbol{Q}}^1 = \boldsymbol{Q}^1 = \alpha^1 \boldsymbol{I} + \beta^1 \boldsymbol{1} \boldsymbol{m}' = \bar{\alpha}^1 \boldsymbol{I} + \bar{\beta}^1 \boldsymbol{1} \boldsymbol{m}'$, satisfying the base case.

**Step II: Inductive Hypothesis.** Assume $t = k$, $\bar{\boldsymbol{Q}}^k = \bar{\alpha}^k \boldsymbol{I} + \bar{\beta}^k \boldsymbol{1} \boldsymbol{m}'$ for $\bar{\alpha}^k = \prod_{\tau=1}^k \alpha^\tau$ and $\bar{\beta}^k = 1 - \bar{\alpha}^k$.

**Step III: Inductive Step.** We prove that $\bar{Q}^{k+1} = \bar{\alpha}^{k+1}I + \bar{\beta}^{k+1}1m'$ for $\bar{\alpha}^{k+1} = \prod_{\tau=1}^{k+1} \alpha^\tau$ and $\bar{\beta}^{k+1} = 1 - \bar{\alpha}^{k+1}$. The detail is shown below:

$$
\begin{aligned}
\bar{Q}^{k+1} &= \bar{Q}^k Q^{k+1} \\
&= (\bar{\alpha}^k I + \bar{\beta}^k 1m') (\alpha^{k+1}I + \beta^{k+1}1m') \\
&= \bar{\alpha}^k \alpha^{k+1}I + (\bar{\alpha}^k \beta^{k+1} + \bar{\beta}^k \alpha^{k+1})1m' + \bar{\beta}^k \beta^{k+1}1m'1m' \\
&= \bar{\alpha}^{k+1}I + \left(\bar{\alpha}^k(1 - \alpha^{k+1}) + (1 - \bar{\alpha}^k)\alpha^{k+1} + (1 - \bar{\alpha}^k)(1 - \alpha^{k+1})\right)1m' \\
&= \bar{\alpha}^{k+1}I + (1 - \bar{\alpha}^{k+1})1m'
\end{aligned}
$$

As $T \to \infty$, $\bar{\alpha}^T \to 0$. Hence $\lim_{T\to\infty} \bar{Q}^T = 1m'$, where all rows are $m'$. Thus, for any base vector $\mathbf{e}_i$, $\lim_{T\to\infty} \bar{Q}^T \mathbf{e}_i = m$.

# B PERMUTATION INVARIANCE AND EXCHANGEABILITY

## B.1 PROOF OF THEOREM 3

**Theorem 3** (Backdoored DiGress is Permutation Invariant). *Let $G^t = (X^t, E^t)$ be an intermediate noised (clean or backdoored) graph, and $\pi(G^t) = (\pi(X^t), \pi(E^t))$ be its permutation. Backdoored DiGress is permutation invariant, i.e., $p_{\theta_B}(\pi(G^t)) = \pi(p_{\theta_B}(G^t))$.*

We need to prove that: i) the neural network building blocks are permutation invariant; and ii) the objection function (i.e., the training loss) is also permutation invariant.

**Proving i):** DiGress uses three types of blocks:

- 1) spectral and structural features (e.g., eigenvalues of the graph Laplacian and cycles in the graph) to improve the network expressivity);

- 2) graph transformer layers (consisting of graph self-attention and fully connected multiple-layer perception);

- 3) layer-normalization.

DiGress proves that these blocks are permutation invariant. Backdoored DiGress uses the same network architecture as DiGress and hence is also permutation invariant.

**Proving ii):** Backdoored DiGress optimizes the cross-entropy loss on clean graphs $\{G = (X, E)\}$ and backdoored graphs $\{G^B = (X_B, E_B)\}$ to learn the model $\theta_B$:

$$
\min_{\theta_B} \mathcal{L}(\{G\}, \{G_B\}; \theta_B)
$$
$$
= \sum_{\{G=(X,E)\}} \left(l_{CE}(X, \hat{p}^X) + l_{CE}(E, \hat{p}^E)\right) + \sum_{\{G^B=(X_B,E_B)\}} \left(l_{CE}(X_B, \hat{p}^{X_B}) + l_{CE}(E_B, \hat{p}^{E_B})\right)
$$

For a clean graph $G$ or a backdoored graph $G_B$, its associated cross-entropy loss can be decomposed to be the sum of the loss of individual nodes and edges. For instance, $l_{CE}(X, \hat{p}^X) = \sum_{1 \le i \le n} l_{CE}(x_i, \hat{p}_i^X)$, $l_{CE}(E_B, \hat{p}^{E_B}) = \sum_{1 \le i,j \le n} l_{CE}(e_{B,ij}, \hat{p}_{B,ij}^E)$.

Hence, the total loss on the clean and backdoored graphs does not change with any node permutation $\pi$. That is,

$$
\mathcal{L}(\{\pi(G)\}, \{\pi(G_B)\}; \theta_B) = \mathcal{L}(\{G\}, \{G_B\}; \theta_B).
$$

## B.2 PROOF OF THEOREM 4

**Theorem 4** (Backdoored DiGress Produces Exchangeable Distributions). *Backdoored DiGress generates graphs with node features $X$ and edges $E$ that satisfy $P(X, E) = P(\pi(X), \pi(E))$ for any permutation $\pi$.*

The proof builds on the result in Xu et al. (2022):

---

**Algorithm 1** Backdoored DiGress Training

---

**Input:** Training graphs $\mathcal{G}_{tr}$, poison rate $p\%$, subgraph trigger $G_s = (\boldsymbol{X}_s, \boldsymbol{E}_s)$, model parameter $\theta_B$, and transition matrices $\{Q_X^t, Q_E^t, Q_{X_B}^t, Q_{E_B}^t\}$.
**Preprocess:** Sample $p\%$ of $\mathcal{G}_{tr}$ and inject $G_s$ to obtain the backdoored graphs $\mathcal{G}_B$; clean graphs $\mathcal{G}_C = \mathcal{G}_{tr} \setminus \mathcal{G}_B$

 1: Sample $G = (\boldsymbol{X}, \boldsymbol{E}) \sim \{\mathcal{G}_B, \mathcal{G}_C\}$
 2: Sample $t \sim Uniform(1, \cdots, T)$
 3: **if** $G \in \mathcal{G}_B$ **then**
 4:    // Sample a noisy backdoored graph
       Sample $G^t \sim (\boldsymbol{X}\bar{Q}_{X_B}^t \odot (1 - \boldsymbol{M}_X) + \boldsymbol{X}_s \odot \boldsymbol{M}_X) \times (\boldsymbol{E}\bar{Q}_{E_B}^t \odot (1 - \boldsymbol{M}_E) + \boldsymbol{E}_s \odot \boldsymbol{M}_E)$
 5: **else**
 6:    Sample $G^t \sim \boldsymbol{X}\bar{\boldsymbol{Q}}_X^t \times \boldsymbol{E}\bar{\boldsymbol{Q}}_E^t$ // Sample a noisy clean graph
 7: **end if**
 8: $\hat{\mathbf{p}}^X, \hat{\mathbf{p}}^E \leftarrow p_{\theta_B}(G^t)$ // Forward pass
 9: // Minimize the cross-entropy loss
    optimizer.step($l_{CE}(\hat{\mathbf{p}}^X, \boldsymbol{X}) + l_{CE}(\hat{\mathbf{p}}^E, \boldsymbol{E})$)

---

**Algorithm 2** Sampling from Backdoored DiGress

---

**Input:** Trained model $p_{\theta_B}$, timestep $T$, marginal distributions $\{\boldsymbol{m}_X^n, \boldsymbol{m}_E^n, \boldsymbol{m}_{X_B}^n, \boldsymbol{m}_{E_B}^n\}$ for all graph sizes $n$.

 1: Sample a graph size $n$ from training data distribution
 2: **if** Generating a clean sample **then**
 3:    Sample $G^T \sim q_X(\boldsymbol{m}_X^n) \times q_E(\boldsymbol{m}_E^n)$
 4: **else**
 5:    Sample $G^T \sim q_X(\boldsymbol{m}_{X_B}^n) \times q_E(\boldsymbol{m}_{E_B}^n)$
 6: **end if**
 7: **for** $t = T$ to 1 **do**
 8:    Forward pass: $\hat{\mathbf{p}}^X, \hat{\mathbf{p}}^E \leftarrow p_{\theta_B}(G^t)$
 9:    Compute node posterior: $p_{\theta_B}(x_i^{t-1}|G^t) \leftarrow \sum_x q(x_i^{t-1}|x_i = x, x^t)\hat{p}_i^X(x)$ $i \in 1, \ldots, n$
10:    Compute edge posterior: $p_{\theta_B}(e_{ij}^{t-1}|G^t) \leftarrow \sum_e q(e_{ij}^{t-1}|eij = e, e_i^t)\hat{p}_{ij}^E(e), i, j \in 1, \ldots, n$
11:    Generate graph from the categorical distribution: $G^{t-1} \sim \prod_i p_{\theta_B}(x_i^{t-1}|G^t) \prod_{i,j} p_{\theta_B}(e_{ij}^{t-1}|G^t)$
12: **end for**
13: **return** $G^0$

---

**Proposition 1** (Xu et al. (2022)). *Let $\mathcal{C}$ be a particle. If,*

*i) a distribution $p(\mathcal{C}^T)$ is invariant under the transformation $T_g$ of a group element $g$, i.e., $p(\mathcal{C}^T) = p(T_g(\mathcal{C}^T))$;*

*ii) the Markov transitions $p(\mathcal{C}^{t-1} \mid \mathcal{C}^t)$ are equivariant, i.e., $p(\mathcal{C}^{t-1} \mid \mathcal{C}^t) = p(T_g(\mathcal{C}^{t-1}) \mid T_g(\mathcal{C}^t))$,*

*then the density $p_\theta(\mathcal{C}^0)$ is also invariant under the transformation $T_g$, i.e., $p_\theta(\mathcal{C}^0) = p_\theta(T_g(\mathcal{C}^0))$.*

We apply Proposition 1 to our setting:

First, the clean or backdoored limit distribution $p(G^T)$ or $p(G_B^T)$ is the product of independent and identical distribution on each node and edge. It is thus permutation invariant and satisfies condition i).

Second, the denoising network $p_{\theta_B}$ in backdoored DiGress is permutation equivariant (Theorem 3). Moreover, the network prediction $\hat{p}_{\theta_B}(G) \to p_{\theta_B}(G^{t-1}|G^t) = \sum_G q(G^{t-1}, G|G^t)\hat{p}_{\theta_B}(G)$ defining the transition probabilities is equivariant to joint permutations of $\hat{p}_{\theta_B}(G)$ and $G^t$, and so to the joint permutations of $\hat{p}_{\theta_B}(G_B)$ and $G_B^t$. Thus, condition ii) is also satisfied.

Together, the backdoored DiGress generated the graph with node features $\boldsymbol{X}$ and edges $\boldsymbol{E}$ that satisfy $P(\boldsymbol{X}, \boldsymbol{E}) = P(\pi(\boldsymbol{X}), \pi(\boldsymbol{E}))$ for any permutation $\pi$, meaning the generated graphs are exchangeable.

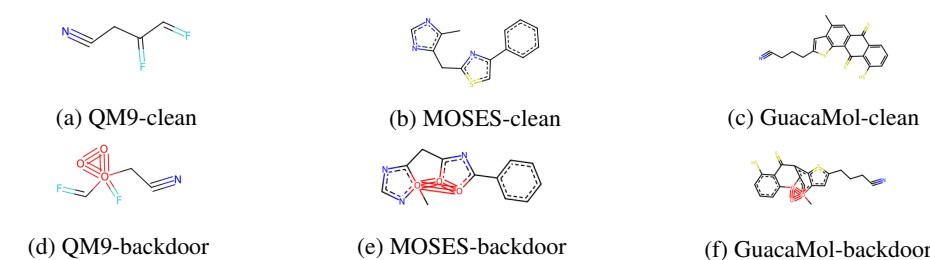

Figure 2: Example clean molecules and their backdoored ones.

## C   RELATED WORK ON GRAPH GENERATIVE MODELS

### C.1   NON-DIFFUSION GRAPH GENERATIVE MODELS

They are classified as *non-autoregressive* and *autoregressive* graph generative models. Non-autoregressive models generate all edges *at once*, and utilize variational autoencoder (VAE) Simonovsky & Komodakis (2018); Ma et al. (2018); Liu et al. (2018); Zahirnia et al. (2022), generative adversarial network (GAN) Maziarka et al. (2020), and normalizing flow (NF) Madhawa et al. (2019); Zang & Wang (2020); Kuznetsov & Polykovskiy (2021) techniques. VAE- and GAN-based methods generate graph edges independently from latent representations, but they face limitations in the size of produced graphs. In contrast, NF-based methods require invertible model architectures to establish a normalized probability distribution, which can introduce complexity and constrain model flexibility.

Autoregressive models build graphs by adding nodes and edges sequentially, using frameworks like NF Shi et al. (2020); Luo et al. (2021), VAE Jin et al. (2018; 2020), and recurrent networks Li et al. (2018); You et al. (2018); Dai et al. (2020). These methods are effective at capturing complex structural patterns and can incorporate constraints during generation, making them superior to non-autoregressive models. However, a notable drawback is their sensitivity to node orderings, which affects training stability and generation performance Vignac et al. (2023).

### C.2   GRAPH DIFFUSION MODELS

Initial attempts for graph generation closely follow diffusion models that rely on continuous Gaussian noise Niu et al. (2020); Jo et al. (2022); Yang et al. (2023). However, continuous noises have no meaningful interpretations for graph data Liu et al. (2023a). To address it, many approach Vignac et al. (2023); Kong et al. (2023); Chen et al. (2023b); Liu et al. (2023a); Li et al. (2024); Gruver et al. (2024); Yi et al. (2024); Xu et al. (2024) propose *discrete* diffusion model tailored to graph data. For instance, DiGress Vignac et al. (2023) extends Liu et al. (2023a) to tailor graph generation with categorical node and edge attributes. By preserving sparsity and structural properties of graphs through a discrete noise model, DiGress effectively captures complex relationships within graphs, particularly crucial for applications like drug discovery and molecule generation, and obtains the SOTA performance. DiGress is also permutation invariant, produces large graphs, and generated graphs are unique and valid, thanks to the exchangeable distribution.

## D   EXPERIMENTS

### D.1   DATASET DESCRIPTION

**QM9**: It is a molecule dataset with 4 distinct elements and 5 bond types. The maximum number of heavy atoms a graph is 9.

**Molecular Sets (MOSES):** It is specially designed to evaluate generative models for molecular graph generation. MOSES consists of molecular structures represented in the SMILES format. The dataset contains 1.9M+ unique molecules derived from the ZINC Clean Leads dataset, ensuring the molecules are drug-like and chemically realistic.

| Dataset | #Epochs | r=0.2 | | | r=0.5 | | | r=1 | | |
|---|---|---|---|---|---|---|---|---|---|---|
| | | ASR | V | U | ASR | V | U | ASR | V | U |
| QM9 | 0 | 100 | 97 | 100 | 100 | 97 | 100 | 100 | 100 | 100 |
| | 10 | 100 | 97 | 100 | 99 | 97 | 100 | 100 | 100 | 100 |
| | 20 | 99 | 98 | 100 | 99 | 98 | 100 | 100 | 100 | 100 |
| | 50 | 98 | 98 | 100 | 99 | 98 | 100 | 99 | 100 | 100 |
| | 100 | 98 | 99 | 100 | 99 | 100 | 100 | 99 | 100 | 100 |
| MOSES | 0 | 90 | 84 | 100 | 87 | 83 | 100 | 86 | 85 | 100 |
| | 10 | 90 | 84 | 100 | 87 | 84 | 100 | 86 | 86 | 100 |
| | 20 | 90 | 85 | 100 | 86 | 83 | 100 | 85 | 84 | 100 |
| | 50 | 88 | 86 | 100 | 85 | 85 | 100 | 82 | 86 | 100 |
| | 100 | 82 | 85 | 100 | 82 | 85 | 100 | 82 | 82 | 100 |
| Guacamol | 0 | 92 | 85 | 100 | 85 | 86 | 100 | 85 | 86 | 100 |
| | 10 | 92 | 84 | 100 | 85 | 86 | 100 | 85 | 85 | 100 |
| | 20 | 90 | 85 | 100 | 84 | 86 | 100 | 83 | 86 | 100 |
| | 50 | 88 | 84 | 100 | 84 | 88 | 100 | 80 | 90 | 100 |
| | 100 | 90 | 86 | 100 | 82 | 87 | 100 | 81 | 92 | 100 |

Table 9: Attack results against finetuning on clean graphs with varying finetuning epochs.

| Dataset | Ratio | r=0.2 | | | r=0.5 | | | r=1 | | |
|---|---|---|---|---|---|---|---|---|---|---|
| | | ASR | V | U | ASR | V | U | ASR | V | U |
| QM9 | 0% | 100 | 97 | 100 | 100 | 97 | 100 | 100 | 100 | 100 |
| | 1% | 99 | 97 | 100 | 99 | 97 | 100 | 100 | 99 | 100 |
| | 2% | 99 | 98 | 100 | 99 | 95 | 100 | 99 | 100 | 100 |
| | 5% | 98 | 97 | 100 | 99 | 92 | 100 | 97 | 100 | 100 |
| | 10% | 98 | 99 | 100 | 99 | 94 | 100 | 98 | 99 | 100 |
| MOSES | 0% | 90 | 84 | 100 | 87 | 83 | 100 | 86 | 85 | 100 |
| | 1% | 89 | 83 | 100 | 86 | 82 | 100 | 86 | 86 | 100 |
| | 2% | 84 | 80 | 100 | 84 | 80 | 100 | 82 | 84 | 100 |
| | 5% | 80 | 81 | 100 | 80 | 83 | 100 | 79 | 81 | 100 |
| | 10% | 77 | 82 | 100 | 75 | 81 | 100 | 77 | 84 | 100 |
| GuacaMol | 0% | 92 | 85 | 100 | 85 | 86 | 100 | 85 | 86 | 100 |
| | 1% | 91 | 85 | 100 | 85 | 87 | 100 | 86 | 85 | 100 |
| | 2% | 89 | 87 | 100 | 84 | 85 | 100 | 83 | 84 | 100 |
| | 5% | 86 | 89 | 100 | 81 | 86 | 100 | 81 | 83 | 100 |
| | 10% | 81 | 84 | 100 | 78 | 87 | 100 | 79 | 84 | 100 |

Table 10: Attack results against finetuning on varying ratios of backdoored graphs mapped to clean limit distribution.

**GuacaMol:** It is a benchmark suite specifically designed for evaluating generative models in molecular discovery. GuacaMol includes a collection of molecules from the ChEMBL database, a large database of bioactive molecules with drug-like properties. The dataset contains 1.3 million drug-like molecules in the SMILES format.

**Training and testing:** On QM9, we use 100k molecules for training, and 13k for evaluating the attack effectiveness and utility. On MOSES, we use 1.58M graphs for training and 176k molecules for testing. On GuacaMol, 200k molecules are used for training and 40k molecules for testing.

### D.2   NETWORK ARCHITECTURE

We use the original DiGress network architecture, which consists of 9 graph transformer layers for QM9, and 12 graph transformer layers for GuacaMol and MOSES.

### D.3   MORE RESULTS

**Visualizing the clean and backdoored graphs generated by the backdoored DiGress.** Figure 3 shows example generated clean graphs, while Figure 4 shows example generated backdoored graphs on the three molecule datasets. We observe that the generated clean graphs are valid, while the backdoored graphs are invalid.

**Comprehensive attack results against backdoor defenses with varying $r$:** Table 9 and Table 10 show the attack results against the two finetuning-based backdoor defenses under different $r$. We see ASR and utility in all epochs or ratios are identical to those without defense. This implies the designed graph backdoor attack is stable w.r.t $r$.

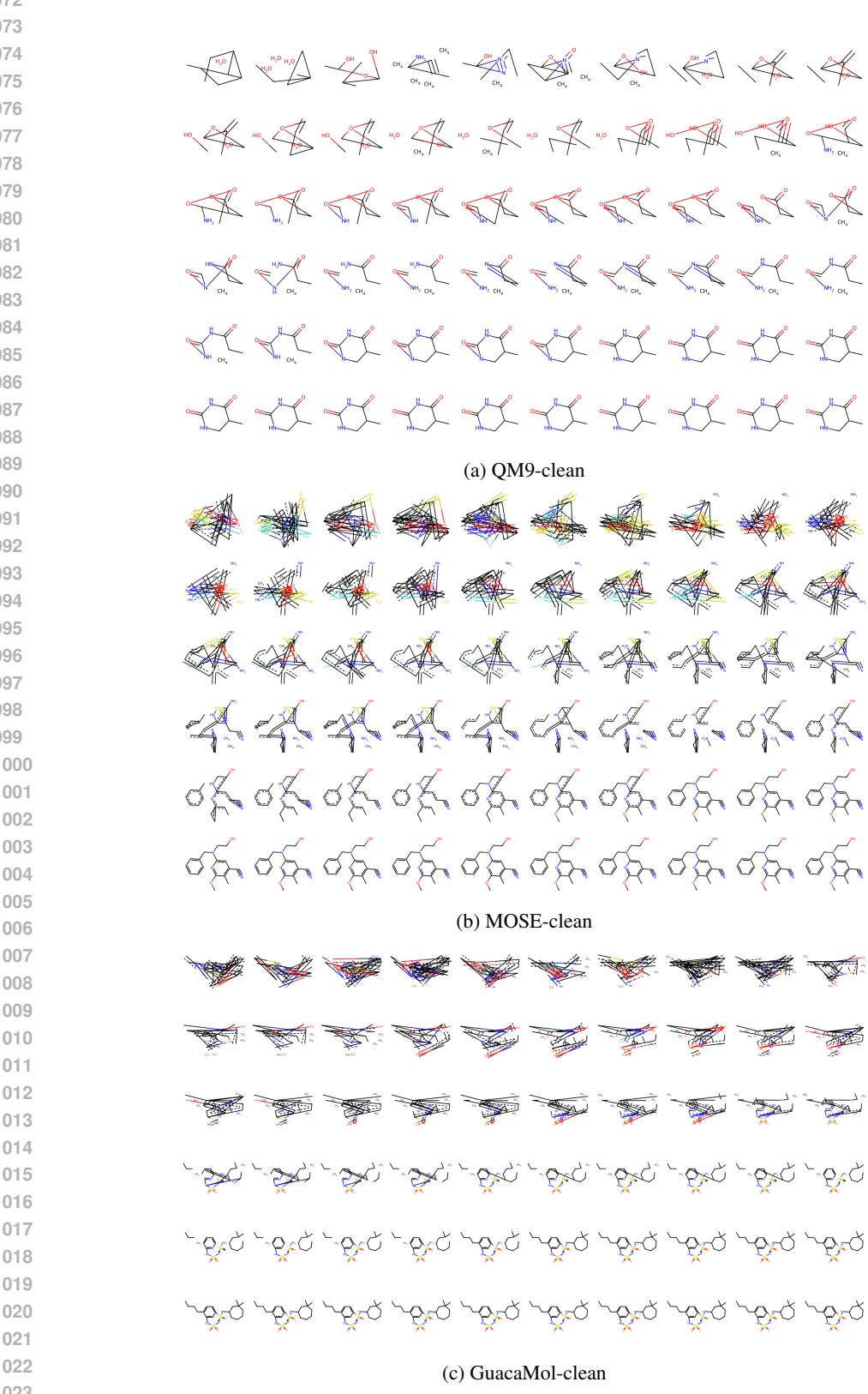

(a) QM9-clean

(b) MOSE-clean

(c) GuacaMol-clean

Figure 3: Example clean graphs generation.

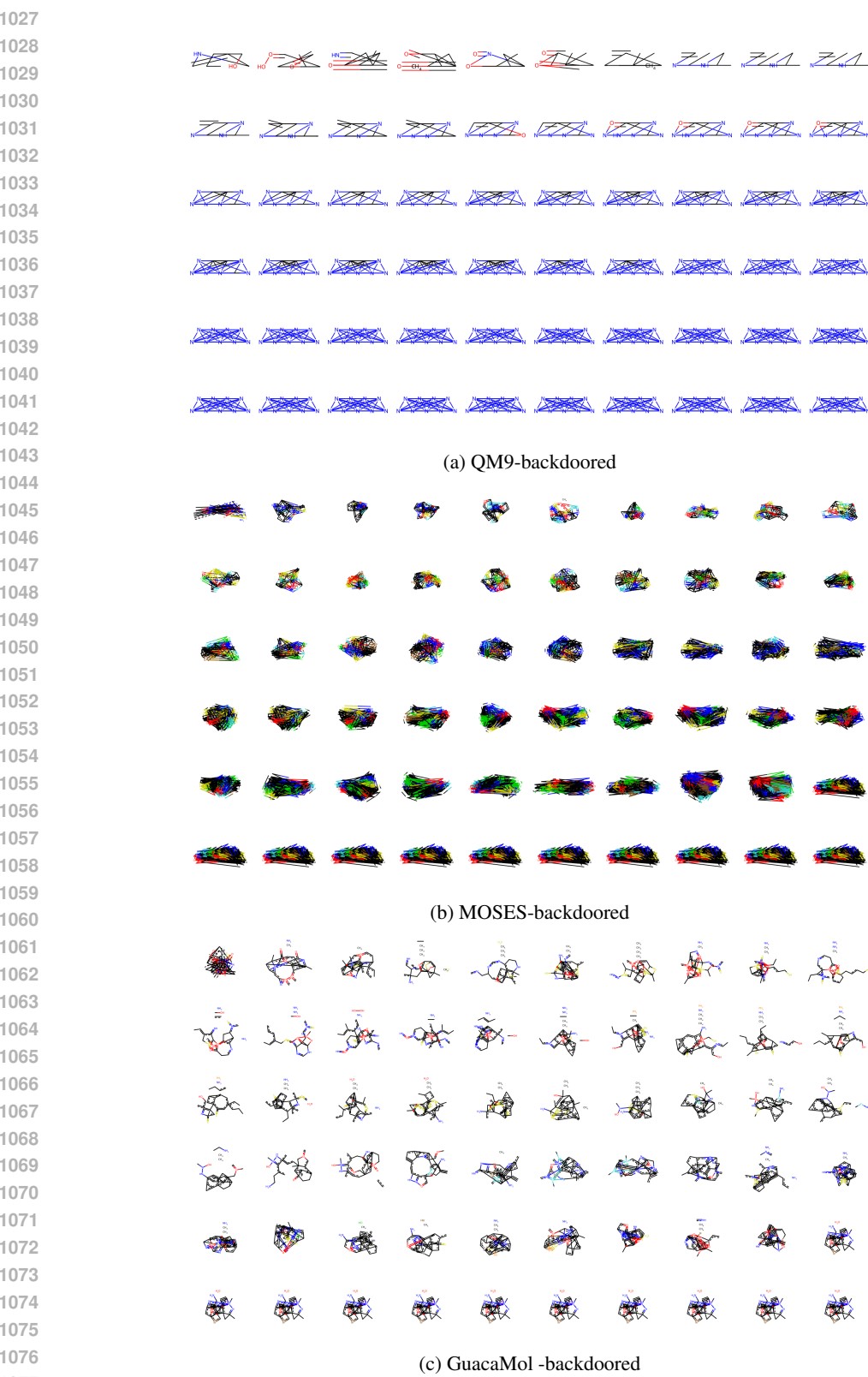

(a) QM9-backdoored

(b) MOSES-backdoored

(c) GuacaMol -backdoored

Figure 4: Example backdoored graphs generation.

