# OpenReview forum: "On the Vulnerability of Discrete Graph Diffusion Models to Backdoor Attacks"
_ICLR.cc/2026/Conference — ICLR 2026 Conference Withdrawn Submission_

### Official Review · Reviewer_sFAf · 2025-10-31

**Soundness:** 2
**Presentation:** 2
**Contribution:** 2
**Rating:** 4
**Confidence:** 3

**Summary:**

The paper explores backdoor attacks on discrete graph diffusion models (DGDMs), claiming to be the first to do so. It formalizes a threat model, designs an attack that maintains utility while embedding a stealthy trigger, and shows empirical results demonstrating its persistence against defenses. The study mainly extends ideas from prior backdoor attacks on diffusion and graph classification models but adapts them to the DGDM context.

**Strengths:**

- Clearly formulated problem setup and systematic explanation of attack procedure, including mathematical grounding.
- Strong empirical evaluation across multiple molecule datasets, including transferability tests and defense evaluations.
- Theoretical discussion of permutation invariance and exchangeability helps maintain consistency with core DGDM properties.

**Weaknesses:**

- While the context (DGDMs) is new, the backbone of the attack is a direct adaptation of established ideas from backdoors in image diffusion and graph classification models. The technical contributions are mainly adaptations rather than fundamentally new algorithms
- Only standard, basic mitigating strategies are tested; there is no deep exploration of advanced, adaptive detection or alternative generative backdoor paradigms.
- The paper asserts security risk based mostly on elevated rates of invalid molecule generation, without strong evidence this translates to actionable harm or abuse potential in realistic, high-value settings.

**Questions:**

1. Why focus on invalid molecule generation, rather than targeting semantically valid but adversarially biased or unsafe outputs?
2. How might more modern graph backdoor detection (beyond simple GED/NLD or finetuning) impact your results?
3. Could your attack generalize to continuous diffusion models, or is it fundamentally tied to discrete Markovian noise?
4. The paper claims “stealthiness” but only reports low graph edit distances. How does this translate to real detectability under more advanced anomaly or structure-aware detection methods?
5. Does the proof of exchangeability (Theorem 2) truly hold when the backdoor alters diffusion transitions asymmetrically across nodes?

---

### Official Review · Reviewer_rqJc · 2025-11-01

**Soundness:** 3
**Presentation:** 3
**Contribution:** 3
**Rating:** 6
**Confidence:** 2

**Summary:**

This paper introduces the first backdoor attack on Discrete Graph Diffusion Models. The authors propose a novel attack that modifies the forward diffusion process to ensure a subgraph trigger is maintained, forcing backdoored graphs to converge to a distinct, attacker-specified limit distribution. The model is then trained to learn both the clean and backdoored diffusion paths. The authors claim their attack is effective with high ASR, preserves model utility, and is persistent against basic finetuning defenses.

**Strengths:**

1. The topic is interesting, with novelty to explore and demonstrate backdoor attacks on DGDMs
2. The attack is not a simple adaptation from other domains. The method of persisting the trigger to create a separate limit distribution is a novel insight specific to diffusion models.
3. The attack is shown to be highly persistent, resisting both finetuning on clean graphs and a form of adversarial finetuning.

**Weaknesses:**

1. The attacker can manipulate the initialization process of diffusion sampling seems to be a strong assumption.
2. The attack creates two separable limit distributions, which seem like a strong, detectable anomaly. It is highly susceptible to standard detection methods (e.g., output clustering, statistical analysis of the latent space), which were not evaluated.

**Questions:**

1. Can this method be extended to a targeted attack (i.e., generating a specific, valid graph)?
2. Why were more sophisticated defenses that could exploit this (e.g., latent space analysis, clustering of generated outputs) not considered in the evaluation?

---

### Official Review · Reviewer_TP3E · 2025-11-01

**Soundness:** 2
**Presentation:** 2
**Contribution:** 2
**Rating:** 2
**Confidence:** 3

**Summary:**

The paper investigates how to inject backdoor triggers into discrete graph diffusion models (DGDMs). The paper argues that existing “know-how” for backdooring graph classification models does not generalise to DGDMs.

**Strengths:**

- backdooring a diffusion model for graphs

**Weaknesses:**

- The threat model is not compelling. The argument that drug designer would inject a trigger to the input to arrive at a target molecule or sample from a backdoor limit distribution to arrive at target molecule makes little sense. Inevitably, the generation process is not perfect/stochastic and the idea would be to explore the design space using a diffusion model and if the generated solution does not yield a result, then one would abandon it or throw away the diffusion model/go to a different provider.

- Further the threat model assumes the attacker has full control over the fine-tuning process and poison data. So, there is no novelty in the threat model compared to existing attacks.

- Showing a possible weakness in a ML pipeline or a model is valuable but this has already been done for diffusion models, for example in BadDiffusion. It is not clear how the attack method in the paper provides a fundamental shift from existing work showing that a diffusion model can be backdoored.

**Questions:**

- What are the technical differences in the proposed attack to those published for demonstrating that a backdoor injection is possible for diffusion models?

- What is the cost of the attack vs success rate vs the backdoored limit distribution distance to the clean limit distribution?

- If one is to receive a diffusion model from a malicious provider, what is the effectiveness of potential counter measures?

- clearly explain how the impact in the main use of a diffusion model for the proposed attack

---

### Official Review · Reviewer_raBn · 2025-11-01

**Soundness:** 2
**Presentation:** 2
**Contribution:** 2
**Rating:** 4
**Confidence:** 4

**Summary:**

This paper presents a backdoor attack on discrete graph diffusion models, specifically targeting DiGress. The attack manipulates the forward diffusion process to make backdoored graphs converge to a different limit distribution than clean graphs, while maintaining a subgraph trigger throughout diffusion. The authors prove that their backdoored model preserves permutation invariance and exchangeability. Experiments on molecular datasets (QM9, MOSES, GuacaMol) demonstrate high attack success rates (87-100%) with minimal impact on clean graph generation quality. The attack shows strong persistence against fine-tuning defenses and transfers across different DGDMs, revealing significant security vulnerabilities in graph generative models for safety-critical applications.

**Strengths:**

1. This work explores the backdoor attack on graph diffusion models, filling a research gap. Compared with backdoor attacks on image diffusion models, the authors clearly articulate the unique challenges in the graph domain like permutation invariance constraint.
2. The paper is theoretically well-founded, and its effectiveness is supported by empirical evaluations.
3. The experiments are generally well-conducted, aside from some issues in result analysis noted below.

**Weaknesses:**

1. The attack is evaluated only on graphs with discrete node features and discrete edges, and the authors do not examine its applicability to the more common case of graphs with discrete edges but continuous node features; this omission significantly limits the attack’s generality.
2. Accessing publicly available pretrained models and fine-tuning them is indeed common practice in NLP and CV. However, in the molecular generation domain, how many real-world systems actually rely on publicly released pretrained models? A more detailed discussion of realistic attack scenarios is needed, or the authors should acknowledge that this is a proof-of-concept study.
3. Using obviously invalid molecules like 'O≡O≡O' 's triggers is unrealistic—routine valency and chemical-validity checks will flag them. Such triggers would be suspicious both at training-data curation and during inference.

**Questions:**

1. The paper mentions that the trigger is injected randomly. How does the injection position influence the attack's success rate?
2. Table 2 shows that as the parameter $r$ decreases, the ASR tends to increase; but intuitively, the greater the divergence between the two distributions, the easier they should be to distinguish, and thus the more effective the attack should be. How can this counterintuitive result be explained?

---

### Note · Authors · 2025-11-28

I have read and agree with the venue's withdrawal policy on behalf of myself and my co-authors.